# A novel assay for improved detection of sputum periostin in patients with asthma

**Junya Ono[1], Masayuki Takai[1], Ayami Kamei[1], Shoichiro Ohta[2], Parameswaran Nair[3], Kenji Izuhara[4], Sven-Erik Dahlén[5], Anna James[5]\*, on behalf of the BIOAIR consortium[¶]**

1 Shino-Test Corporation Ltd., Sagamihara, Japan, 2 Department of Laboratory Medicine, Saga Medical School, Saga, Japan, 3 Department of Medicine, Division of Respirology, McMaster University and Firestone Institute for Respiratory Health, St Joseph's Healthcare, Hamilton, Ontario, Canada, 4 Division of Medical Biochemistry, Department of Biomolecular Sciences, Saga Medical School, Saga, Japan, 5 Experimental Asthma and Allergy Research, Institute of Environmental Medicine, Karolinska Institutet, Stockholm, Sweden

¶ Membership of the BIOAIR Consortium is provided in Acknowledgments.
* Anna.James@ki.se

**Data Availability Statement:** All relevant data are within the paper and its Supporting Information files.

**Funding:** We would like to thank the following funding sources for financial support: the Fifth

## Abstract

### Background

Serum periostin associates with type-2 inflammation in asthmatic airways, but also reflects whole body periostin levels originating from multiple sources. Less is known about sputum periostin as a biomarker in asthma as detection levels are low using currently available periostin assays. We aimed to investigate detection of sputum periostin using ELISA assays targeting different periostin epitopes and relate levels to clinical characteristics.

### Methods

Two ELISA systems were developed using antibodies detecting whole periostin or cleavage products, the molecular weight and amino acid sequences of which were confirmed. The ELISA assays were applied to sputum from 80 patients with mild-to-moderate and severe asthma enrolled in the European, multi-center study BIOAIR. Results were related to clinical characteristics.

### Results

Sputum was found to contain smaller periostin fragments, possibly due to proteolytic cleavage at a C-terminal site. Comparing ELISA methodology using antibodies against cleaved versus whole periostin revealed detectable levels in 90% versus 44% of sputum samples respectively. Sputum periostin showed associations with blood and sputum eosinophils. Furthermore, sputum, but not serum, periostin correlated with reduced lung function and sputum IL-13 and was reduced by oral corticosteroid treatment.

### Conclusions

We present an ELISA method for improved analysis of sputum periostin by detecting cleavage products of the periostin protein. Using this assay, sputum periostin was detectable and associated with more disease-relevant parameters in asthma than serum periostin. Sputum

Framework Programme of the European Union (contract number QLG1-CT-2000-01185), Hjärt-Lungfonden (grant numbers 20140533, 20170450, 20180658, 20200778), Vetenskapsrådet (grant numbers 2014-26826, 2018-02851), Astma och Allergiförbundet (grant number F2017-0024), Stiftelsen för Strategisk Forskning (grant number RB13-0196), Stockholm Läns Landsting (ALF grant numbers 1411-1372, 2017-1341, 2018-1157 and 2019-1054), The ChAMP (Centre for Allergy Research Highlights Markers of Asthma Phenotype) consortium which is funded by the Karolinska Institutet, AstraZeneca & Science for Life Laboratory Joint Research Collaboration (grant number 4-665/2013) and Vårdal Stiftelsen (grant number 2014-0118), as well as The Frederick E. Hargreave Teva Innovation Chair in Airway Diseases. The funders had no role in study design, data collection and analysis, decision to publish, or preparation of the manuscript.

periostin is worth considering as a phenotype-specific biomarker in asthma as its proximity to the airways may eliminate some of the confounding factors known to affect serum periostin.

## Introduction

In recent years, periostin as measured in serum or plasma, has emerged as a biomarker of type-2 inflammation in asthma due to its marked up-regulation by IL-13 in airway epithelial cells and fibroblasts [1, 2], alongside significant relationships with other surrogate markers of type-2 inflammation including blood and sputum eosinophils and exhaled nitric oxide [3]. Very briefly, the role of periostin, a matricellular protein, in asthma is believed to involve elevated production by structural cells in response to IL-13 secreted by activated T-helper 2 cells or type 2 innate lymphoid cells, which in turn can lead to increased airway fibrosis [4].

Despite consistent associations with eosinophilic inflammation, recent investigations suggest that circulating periostin may not be the most sensitive and specific indicator of type-2 asthma [5]. One factor complicating the utility of circulating periostin as a biomarker in asthma may be that its levels are affected by periostin originating from sources outside the lung. Indeed, periostin levels are known to be affected by confounding factors such as BMI, age and bone growth [3, 6]. Hence there is good reason to further investigate the utility of periostin originating from the diseased organ itself and one may hypothesize that sputum periostin levels, originating from the airways, are potentially more reflective of an involvement in asthma pathogenesis than periostin in the blood.

There has been uncertainty concerning whether periostin is present in airway secretions as periostin release from cultured airway epithelial cells has been shown to occur in a basolateral direction and not into the airway lumen [7]. However, a small number of publications do describe the presence of periostin in sputum, although levels are several-fold lower than in the blood and more difficult to assess [8, 9]. Detection may be complicated by methodological factors including the fact that sputum supernatants tend to be diluted, and to contain proteases that can affect proteins of interest.

It was therefore our aim to comprehensively investigate detection levels of sputum periostin using antibodies targeting different epitopes of the periostin protein. As well as investigating the size and amino acid sequences of detectable periostin products in serum and sputum, we compared two different ELISA systems consisting of different antibody pairs to enable detection of either whole or cleaved periostin. The two assays were applied to sputum and serum samples from 80 well-characterized patients with asthma, both mild-to-moderate and severe, from the European multicentre study BIOAIR [10], where associations with relevant clinical characteristics, including other markers of type 2 inflammation, as well as corticosteroid therapy, could be investigated. Furthermore, we were able to investigate relationships between sputum periostin and type 2 cytokines (IL-4 and IL-13) in a smaller, unrelated subset of 30 patients with asthma.

## Materials and methods

### Periostin proteins and antibodies

We prepared recombinant periostin proteins using Drosophila S2 cells [1] and purified serum periostin as previously reported [11, 12]. We then generated eight different monoclonal

antibodies (mAbs) against periostin by immunizing rats or mice with recombinant human periostin, also as previously described [12]. The epitope mapping of the antibodies SS16A, SS17B, SS18A, SS19A, SS19C, SS19D, SS20A and SS21A [11], is shown in Fig 1B. Immunoprecipitation and Western blotting were both performed as previously described [11–13].

## Development of ELISAs detecting whole and cleaved periostin

Two ELISA systems were constructed for measuring periostin, Assay A and Assay B, as shown in Fig 1D. Both Assay A and Assay B used SS18A as the capture Ab. Assay A used SS17B as the secondary Ab, whereas Assay B used SS19C. To increase the sensitivity of periostin detection in this study, we modified the original ELISA system [14] using the biotin-streptavidin method [15]. Briefly, we used biotin-labelled secondary Abs, SS17B at 50 ng/mL or SS19C at 200 ng/mL followed by peroxidase-labelled streptavidin (concentration 1/15,000) (Stereospecific Detection Technologies). We calculated periostin concentrations in the serum or sputum using recombinant periostin proteins. Samples were diluted and measured at 1:5000 for serum or 1:50 to 1:100 for sputum. All samples were analyzed in duplicate and average values are reported. Assay validation was carried out for both Assay A and Assay B, and all assay relevant information including assay variability, detection limits, interference testing and calibration curves are shown in supporting S1 and S2 Tables, and S1 Fig. The measurement of serum periostin has been extensively reported using Assay A, which was the method used for analysis of serum in the current investigation [4].

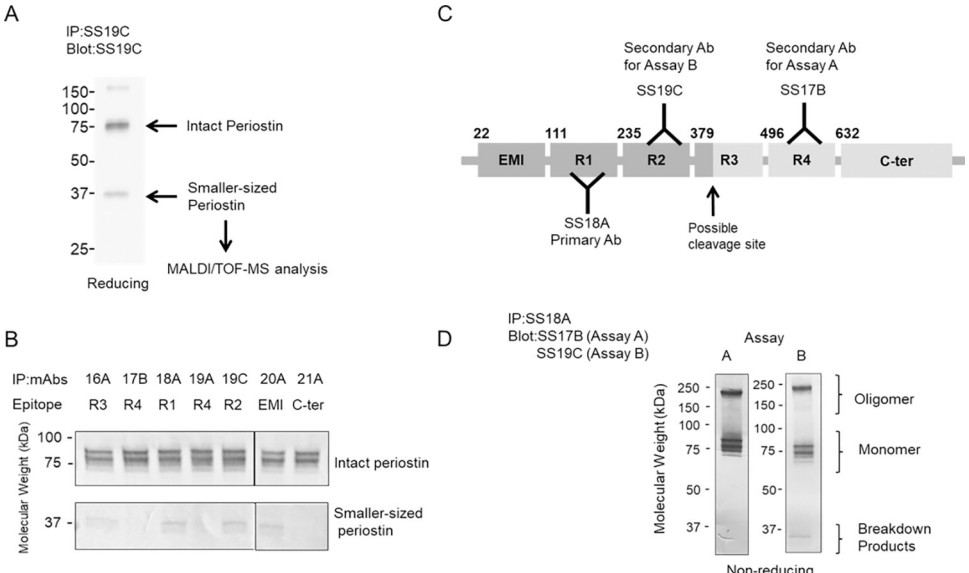

**Fig 1. Identification of smaller periostin fragments.** (A) Immunoprecipitated proteins by SS19C blotted by SS19C. The arrows indicate intact periostin and smaller-sized periostin. The smaller product was cut from the gel and subjected to MALDI/TOF-MS. (B) Reactivities of cleaved periostin against anti-periostin mAbs. Periostin purified from human serum was immuno-depleted by SS16A, SS17B, SS18A, SS19A SS19C, SS20A, or SS21A. Western blotting of the immunoprecipitate by the indicated mAbs, followed by blotting with SS19C under reducing conditions is depicted. (C) Antibody binding sites in the original (Assay A) and novel (Assay B) periostin ELISA systems. The arrow indicates the possible cleavage site of the smaller-sized product of periostin. Dark gray parts indicate the composition of the smaller-sized periostin. The capturing mAb (SS18A) and secondary mAbs (SS17B and SS19C) used in Assay A and Assay B are shown. (D) Reactivities of periostin for Assay A and B showing periostin oligomers, monomers and smaller-sized periostin fragments.

## MALDI/TOF-MS

Serum periostin and recombinant periostin proteins contain both monomeric and oligomeric forms, the latter formed by intermolecular disulfide bonds [11, 12, 16, 17]. Certain antibodies against serum periostin can detect a smaller product at 37 kDa (Fig 1A and 1D). This 37 kDa product was detected by mAbs recognizing EMI, R1, R2 or R3 regions, but not R4 or C-terminal regions of the periostin protein (Fig 1B and 1C). MALDI/TOF-MS analysis was used to confirm that the smaller periostin product is composed of EMI, R1, R2, and the first part of the R3 domain, but not R4 and C-terminal domains (S2 Fig).

Briefly, purified periostin obtained from human serum was separated by SDS-PAGE and the band corresponding to 37 kDa was excised and stained using a silver-staining kit (ATTO Corporation, Tokyo). We digested proteins in the gel piece with 12.5 mg/L trypsin (Promega, Madison, WI) in 25mM ammonium hydrogen carbonate pH 7.8. Then, we tandemly subjected the samples to reverse-phase liquid chromatography (Nano LC System DiNa, Techno Alpha, Tokyo, Japan) and time-of-flight mass spectrometry (4800 Plus MALDI TOF/TOFTM, AB SCIEX, Framingham, MA). We used the Data Explorer software tool (AB SCIEX) to identify proteins based on m/z data of digested peptides and fragmentation of selected peptides. Selected spectra were submitted to Mascot to enable search in the SwissProt database for MS ions.

## Digestion of purified serum periostin by MMP-7

The C-terminal amino acid sequence detected by the MALDI/TOF-MS analysis, LVAQL, is a probable cleavage site for MMP-7, a protease known to be detectable in induced sputum [18]. To compare the specificities of Assay A and Assay B for cleaved periostin, we incubated purified serum periostin with MMP-7 to enable digestion, before ELISA analysis. We added 1 μg of recombinant human matrix metalloproteinase-7 (MMP-7) (R&D systems, Minneapolis, MN, USA) to 500 ng of purified serum periostin in 1 mL of reaction buffer (200 mM NaCl, 50 mM Tris, pH 7.6, 2 mM CaCl2) with or without 1 mM EDTA, and incubated these together for up to 6 hours at 37˚C. Digestion of purified serum periostin with MMP-7 produced a cleavage product that migrated at around 37 kDa, similar to the smaller-sized product existing in serum (S3A Fig). Further, this procedure decreased the reactivity of Assay A, but did not affect Assay B (S3B Fig). Addition of EDTA to the reaction buffer containing MMP-7 inhibited the digestion of periostin and prevented the reduced response of Assay A, confirming the different sensitivities of the two ELISAs for protease-digested periostin.

The original ELISA system we developed for periostin utilizes SS18A and SS17B antibodies (Assay A), which recognize R1 and R4 domains respectively, and cannot detect the smaller-sized product (Fig 1D). Therefore, we established a novel ELISA system to detect the smaller-sized product of periostin using SS18A and SS19C recognizing R1 and R2 domains, respectively (Assay B). Regarding the detection of serum periostin, there was a strong correlation between the values reported by the two assays, demonstrated in supporting S4 Fig.

## Patient samples

Sputum samples were obtained from 80 patients with asthma, 40 mild-to-moderate and 40 severe, taking part in the European multicenter study BIOAIR, which has been described in detail previously [19]. Patients taking part in this study underwent a 2-week double-blind placebo-controlled oral steroid intervention, consisting of a standard course of prednisolone (0.5 mg/kg of body weight/day) added to regular treatment. Furthermore, patients attended up to 6 scheduled clinic visits over the one-year study period. Subject characteristics at study entry are shown in Table 1. The BIOAIR study was approved by the local ethics committee of each participating center, and all participants provided written, informed consent.

**Table 1. Clinical characteristics of BIOAIR study participants.**

| | Mild-to-moderate Asthma | | Severe Asthma | | Mann-Whitney Test (* or Chi$^2$) |
|---|---|---|---|---|---|
| | Median (IQR) | n | Median (IQR) | N | P |
| Sex (% female) | 60% | 40 | 60% | 40 | >0.9999* |
| Age (years) | 42 (33–52) | 40 | 54 (44–60) | 40 | 0.001 |
| BMI (kg/m$^2$) | 24.6 (22.0–27.4) | 40 | 27.4 (24.8–30.7) | 40 | 0.83 |
| ICS (μg/day, beclomethasone eq.) | 500 (400–640) | 40 | 1500 (1000–1950) | 40 | <0.0001 |
| OCS (% taking) | 0% | 40 | 32.5% | 40 | <0.0001* |
| FEV$_1$ (% predicted) | 87.4 (74.4–97.0) | 37 | 72.5 (54.9–88.9) | 40 | 0.007 |
| FVC (% predicted) | 106.3 (98.7–113.5) | 37 | 90.6 (79.0–111.4) | 40 | 0.007 |
| FEV$_1$/FVC (ratio) | 0.69 (0.61–0.76) | 37 | 0.66 (0.57–0.74) | 40 | 0.11 |
| Bronchial reversibility (%) | 9.7 (7.5–13.7) | 40 | 8.1 (2.1–13.4) | 40 | 0.12 |
| FeNO (ppb) | 34.3 (24.1–51.6) | 22 | 41.3 (18.5–93.9) | 18 | 0.51 |
| IgE (KU/l) | 109 (33–254) | 36 | 146 (51–306) | 40 | 0.60 |
| Atopy (% with positive skin prick test) | 62% | 37 | 55% | 40 | 0.52* |
| Sputum periostin Assay A (ng/ml) | 0.00 (0.00–0.35) | 40 | 0.00 (0.00–0.58) | 40 | 0.59 |
| Sputum periostin Assay B (ng/ml) | 0.30 (0.20–0.88) | 40 | 0.70 (0.10–1.95) | 40 | 0.17 |
| Blood eosinophils (x10$^3$ cells/μl) | 0.25 (0.16–0.37) | 39 | 0.31 (0.20–0.58) | 39 | 0.18 |
| Sputum eosinophils (%) | 1.2 (0.4–6.2) | 40 | 5.6 (1.0–27.3) | 36 | 0.014 |
| Sputum neutrophils (%) | 42.7 (18.0–68.0) | 40 | 42.2 (26.1–68.4) | 36 | 0.75 |
| Serum periostin (ng/ml) | 84 (69–102) | 35 | 84 (74–109) | 39 | 0.60 |

Abbreviations: BMI, body mass index; FEV1, forced expiratory volume in one second; FVC, forced vital capacity; IgE, immunoglobulin E; FeNO, fractional exhaled nitric oxide.

In order to examine the possible relationship between sputum periostin and type 2 cytokines, additional analyses were performed in a separate group (cohort 2) of 30 patients with asthma, the characteristics of whom are shown in supporting S3 Table. These patients were taking part in clinical investigations into asthma and COPD in sputum cells as approved by the Hamilton Integrated Research Ethics Board, ON, Canada (Date: 14th November 2018, Reference: 12–3716), and again, all participants provided written, informed consent.

## Sputum processing

Sputum induction was carried out as described previously [20]. Briefly, sputum was dispersed in a DTT/PBS solution where selected sputum plugs were treated with 4 volumes of 6.5mM DTT for 15 minutes and then diluted with PBS to a final concentration of 3.25mM. Periostin levels were measured in sputum supernatant fractions.

## Statistical analysis

Unless otherwise stated, between group differences were examined by non-parametric Kruskal-Wallis test (3 or more groups) or Mann-Whitney U-test (2 groups). Associations between sputum or serum periostin levels with clinical characteristics were examined by Spearman Rank correlation. The Chi$^2$ test was used to compare the difference in proportions of males/females, those taking oral corticosteroids and those with a positive skin prick test between the different asthma groups. All analyses were performed using GraphPad Prism statistical software (GraphPad Software, San Diego, CA) and p-value of <0.05 was considered significant.

## Results

### Application of ELISA Assay A and Assay B to sputum from patients with asthma

Sputum samples from patients with asthma were analyzed using the two periostin ELISAs, Assay A and Assay B. The clinical characteristics and periostin levels of these patients are shown in Table 1. Sputum periostin was detectable in 44% of all samples analyzed using Assay A, and 90% of all samples using Assay B. Positive detection levels were similar in the mild-to-moderate (43% assay A, 88% assay B) and severe asthma (45% assay A, 93% assay B) groups. Similar detection levels were observed in the same patients at three further follow-up visits, supporting S4 Table.

As serum periostin has previously been shown to associate with measures of type-2 inflammation, we compared relationships between periostin in sputum and serum, with relevant clinical characteristics in patients with asthma, Table 2. Overall, results showed that sputum periostin associated with more clinical characteristics and surrogate markers of type 2 inflammation than periostin in serum, including negative correlations with measures of lung function. Sputum and serum periostin levels were found to correlate with each other and both showed significant positive correlations with blood and sputum eosinophil numbers. Scatter plots of significant correlations between sputum periostin and clinical characteristics from Table 2 are shown in supporting S5 Fig.

As sputum cytokines were not investigated as part of the BIOAIR study, we carried out an additional investigation into the relationship between sputum periostin with sputum IL-4 and IL-13 in a separate cohort of patients with asthma. Results showed a significant correlation between sputum, but not serum, periostin with IL-13 (Spearman r = 0.548, p = 0.004), S5 Table. In line with this association, dividing patients into those with sputum periostin above and below the group median, sputum IL-13 levels were found to be significantly greater in those with higher sputum periostin (Fig 2).

**Table 2.  Correlations between serum and sputum periostin and clinical characteristics in patients with asthma from BIOAIR study.**

| | Sputum periostin (Assay A) | | Sputum periostin (Assay B) | | Serum periostin | |
|---|---|---|---|---|---|---|
| *Correlation vs* | *Spearman r* | *P* | *Spearman r* | *p* | *Spearman r* | *P* |
| Sputum periostin (Assay A) | | | **0.694** | **<0.0001** | **0.315** | **0.006** |
| Sputum periostin (Assay B) | **0.694** | **<0.0001** | | | **0.321** | **0.005** |
| Serum periostin | **0.315** | **0.006** | **0.321** | **0.005** | | |
| Sputum eosinophils | **0.345** | **0.002** | **0.353** | **0.002** | **0.413** | **0.0004** |
| Blood eosinophils | **0.344** | **0.002** | **0.498** | **<0.0001** | **0.367** | **0.002** |
| FeNO | 0.301 | 0.059 | **0.311** | **0.050** | 0.094 | 0.582 |
| Bronchial reversibility | -0.006 | 0.959 | 0.024 | 0.832 | -0.083 | 0.480 |
| Total IgE | 0.131 | 0.257 | 0.196 | 0.087 | 0.211 | 0.075 |
| $FEV_1$ | **-0.234** | **0.040** | **-0.455** | **<0.0001** | -0.178 | 0.135 |
| FVC | 0.077 | 0.508 | -0.133 | 0.250 | -0.095 | 0.427 |
| $FEV_1$/FVC | **-0.487** | **<0.0001** | **-0.663** | **<0.0001** | -0.226 | 0.056 |
| ICS | 0.037 | 0.742 | 0.105 | 0.352 | 0.008 | 0.949 |
| BMI | -0.105 | 0.354 | -0.015 | 0.897 | -0.135 | 0.253 |
| Age | 0.036 | 0.752 | 0.168 | 0.137 | -0.080 | 0.499 |

Spearman rank correlations examined in serum and sputum samples obtained from asthma patients only from the BIOAIR study. Abbreviations: BMI, body mass index; FEV1, forced expiratory volume in one second; FVC, forced vital capacity; IgE, immunoglobulin E; FeNO, fractional exhaled nitric oxide; IL, interleukin.

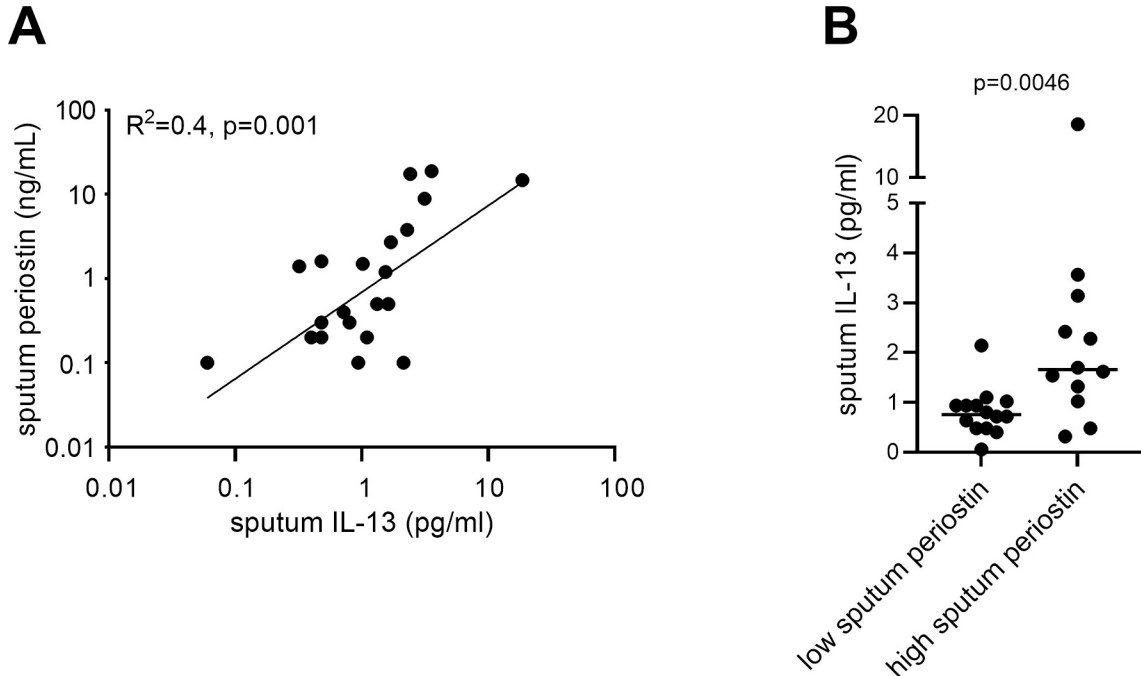

**Fig 2. Association between sputum IL-13 and sputum periostin.** The relationship between sputum IL-13 and sputum periostin (assay B) was examined in an additional, independent cohort of asthma patients. Panel A shows a scatterplot of sputum IL-13 and periostin levels analysed by linear regression ($R^2$ = 0.4, p = 0.0013). The Spearman r for this analysis was 0.59, p = 0.0031. Panel B shows sputum IL-13 levels in patients with sputum periostin $<$ or $\geq$ the group median, termed low sputum periostin and high sputum periostin. Individual data points are shown with line at group median.

Patients with asthma were also subdivided according to sputum inflammatory profile, Fig 3. Dividing patients with asthma into those with eosinophilic and non-eosinophilic asthma based on the cut-off of 3% sputum eosinophils was associated with greater levels of serum and sputum periostin in the eosinophil-high group. Dividing patients according to the four sputum inflammatory profiles paucigranulocytic, neutrophilic, eosinophilic and mixed granulocytic revealed significant differences in sputum, but not serum periostin.

## Effect or oral corticosteroid treatment on serum and sputum periostin levels

The effect of a two-week, placebo-controlled oral steroid intervention was examined in a subset of BIOAIR patients. Significant reductions in sputum periostin were observed following corticosteroid treatment, whereas the change in serum periostin did not reach statistical significance, Fig 4.

## Existence of the cleaved product of periostin in sputum from asthma patients

Given that the novel periostin ELISA system, Assay B, can detect more periostin in sputum than the original periostin ELISA system, Assay A, we hypothesized that sputum from asthma patients contains the smaller-sized periostin product, most likely due to cleavage by a protease. To explore this possibility, we carried out Western blotting on sputum samples from mild and severe asthma patients, to examine whether the cleaved product could be detected. Neither the monomeric form, nor the cleaved product could be detected in sputum samples derived from

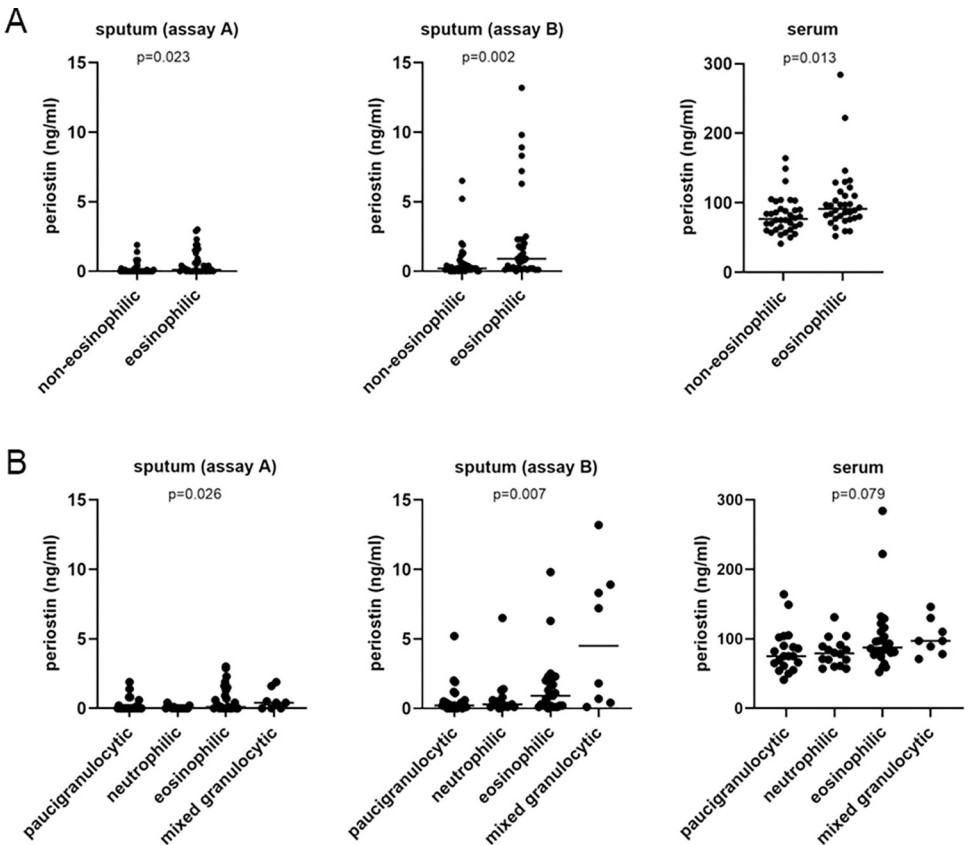

**Fig 3. Periostin levels in serum and sputum samples subdivided according to sputum inflammatory profile.**
Periostin levels were measured in serum and sputum (Assay A and Assay B) in BIOAIR patients and subdivided
according to A) sputum eosinophils (eosinophilic = 3% or greater eosinophils, non-eosinophilic below 3% eosinophils)
and B) sputum inflammatory profile. Briefly, the four inflammatory profiles were defined as follows:
neutrophilic = 61% or greater sputum neutrophils (eosinophils under 3%), eosinophilic = 3% or greater eosinophils
(neutrophils under 61%), paucigranulocytic = less than 61% neutrophils and less than 3% eosinophils, and mixed
granulocytic = 61% or greater neutrophils and 3% or greater eosinophils. 33% of patients were paucigranulocytic, 21%
were neutrophilic, 35.5% were eosinophilic and 10.5% were mixed granulocytic.

three mild asthma patients (#1 - #3), compatible with the presence of very low levels of sputum
periostin as measured by both Assay A and Assay B (Fig 5). Sputum samples derived from
three severe asthma patients (#4 - #6) with high sputum periostin levels measured by Assay A
and Assay B showed clear bands corresponding to both the monomeric form and the cleaved
product. Moreover, sputum samples from four severe asthma patients (#7 - #10) showing high
sputum periostin levels measured by Assay B, but not by Assay A, showed clear bands corre-
sponding to the cleaved product, but not the monomeric form. The size of the smaller perios-
tin product derived from severe asthma patients (#5 - #9) was in the region of 37 kDa, similar
to that of the cleaved periostin product shown in Fig 1A.

## Discussion

In the current investigation we present an ELISA system capable of improved detection of
periostin in sputum, and provide evidence that this is due to the presence of cleaved periostin
fragments in sputum that can be detected by periostin antibodies binding specific protein
regions only. The ELISA system with improved detection levels consists of an antibody pair
(SS18A and SS19C) detecting both whole (oligomers, monomers) and cleaved periostin

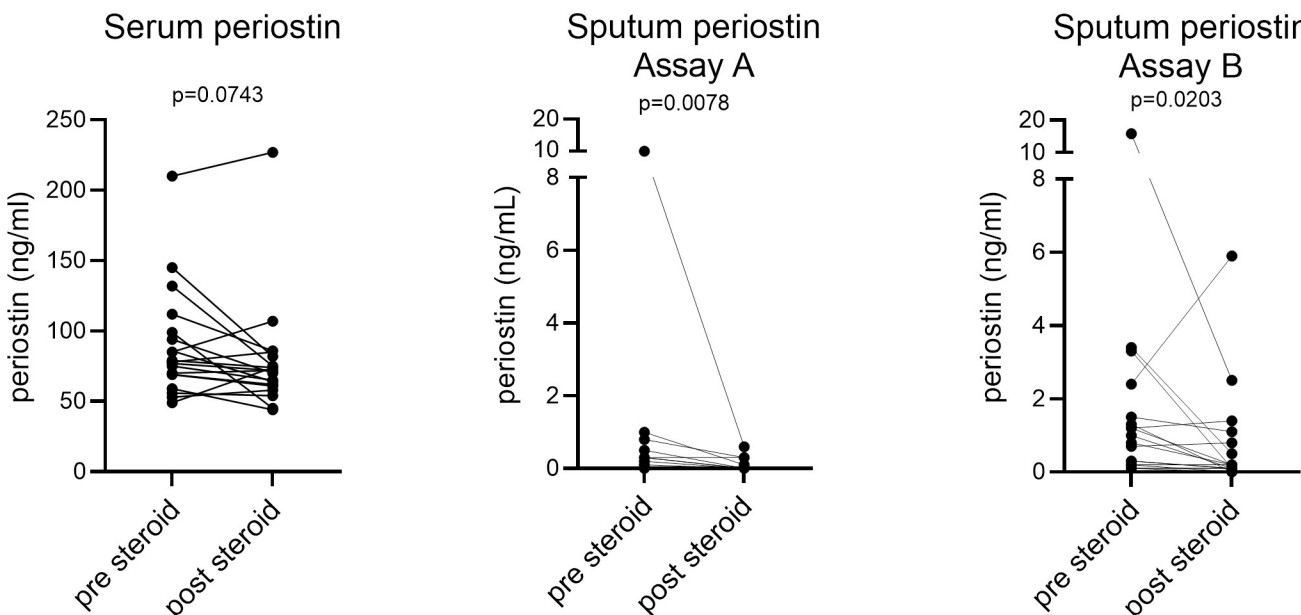

**Fig 4. Effect of oral corticosteroid treatment on serum and sputum periostin.** Periostin levels in matched serum and sputum samples were measured before and after 2 weeks of treatment with oral prednisolone. Paired comparisons by Wilcoxon matched-pairs signed rank test were possible for n = 20 patients with mild-to-moderate and severe asthma from the BIOAIR study.

products. In the current investigation, initially aimed at method development, it was also possible to compare serum and sputum periostin measurements in clinical cohort of patients with asthma. Generally, more significant associations with measures of type 2 inflammation and lung function were apparent for sputum periostin compared to serum periostin.

We provide evidence that the ELISA system with improved detection of sputum periostin (Assay B) is due to this method picking up cleaved periostin fragments, which are present predominantly in sputum and to a lesser extent in serum. The amino acid sequence of cleaved periostin fragments corresponds to EMI, R1, and R2 domains of the protein structure, and the

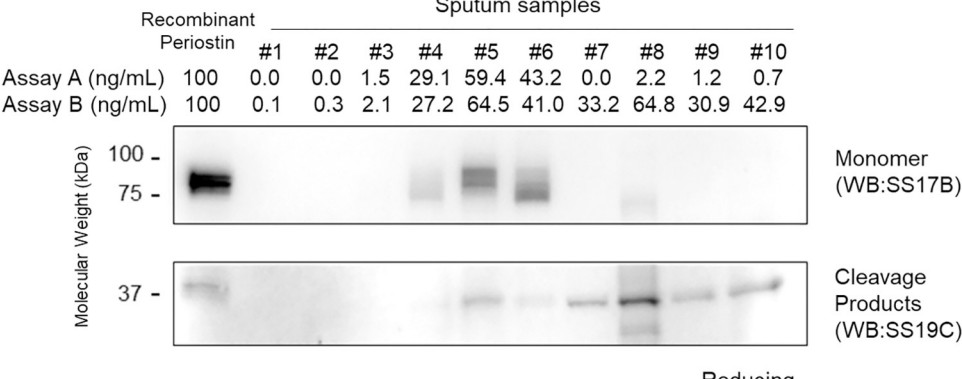

**Fig 5. Evidence of intact and cleaved periostin products in sputum from asthma patients.** Western blot showing intact and/or cleaved periostin products as well as estimated periostin values obtained by ELISA. The results of Assay A or Assay B in sputum samples from 10 asthma patients are depicted. In the Western blotting image, the monomeric periostin represents the reaction with the SS17B antibody and the cleaved product with the SS19C antibody.

first part of the R3 domain, which are only detected by a specific ELISA antibody combination. Furthermore, a potential cleavage site for MMP-7 was identified in the periostin protein, and digestion with this enzyme resulted in a cleavage product migrating at around 37 kDa, similar to the smaller-sized product identified. MMP-7, or matrilysin, is secreted by airway epithelial cells and macrophages and is detectable in samples of induced sputum [18]. As MMP-7 digestion decreased the reactivity of Assay A, but had no effect on Assay B we confirm the different sensitivities of the two ELISAs for protease-digested periostin, and provide further evidence that Assay B shows greater detection of periostin cleavage products.

There is a need for improved methodology to measure sputum periostin. Airway secretions are a valuable source of biomarkers in asthma but can also be a complex matrix to work with as they are often diluted several-fold, have additives such as DTT (sputolysin) to facilitate processing, and may contain active proteases [21]. In fact, it is well known that sputum from patients with asthma may have high proteolytic activity due to the secretion of proteases by activated granulocytes, which in turn can affect protein detection, e.g. cytokines [22]. Initial investigations of sputum periostin as a biomarker in asthma have all described relatively low detection levels in this matrix, sometimes below the lower detection limits of the assays used [8, 9, 23]. Nevertheless, where periostin has been detected in induced sputum or exhaled breath condensate (EBC) there does seem to be an association with type-2 driven asthma as sputum periostin levels have been shown to correlate with airway eosinophils and/or to be elevated in patients with eosinophil-dominant asthma [8, 9, 23, 24]. These findings support further investigations of periostin as a biomarker in sputum, and the development of improved methodology to do so.

The difficulty in obtaining standardized sputum samples is one of the drawbacks of the current investigation, and all others investigating sputum as a source of biomarkers in asthma, as it may affect the clinical utility of sputum periostin measurements. However, efforts are ongoing to improve standardization, both the European Respiratory Society and American Thoracic Society have published guidelines to help standardize methodology, and this problem may be reduced or overcome in the future.

As periostin has been shown to be released by cultured epithelial cells in a basolateral direction only [7], the source of periostin in airway secretions may be debated. Periostin is present in the plasma and there may be plasma leakage into the airway lumen through a damaged epithelial layer. Despite initial observations that periostin is released from epithelial cells basolaterally, another possibility is that periostin is actively secreted into the airway lumen by a currently unknown mechanism. This would be in line with a role for periostin (and IL-13) in mucosal immunity as suggested by the presence of excess periostin and IL-13, in tear fluid from patients with severe allergic conjunctivitis [25]. Fibroblasts in conjunctival tissues, but not corneal cells, produce periostin suggestive of a system for periostin secretion from stroma to lumen [25]. Either way, the source of periostin deserves further investigation as one may hypothesize that if sputum periostin originates primarily from the airways, it may have more, or different, usefulness as a biomarker by more closely reflecting airway inflammation, mucus secretion or fibrosis, and its levels could potentially be less affected by known confounding factors such as e.g. bone growth, BMI and age [3, 6].

Interestingly, there is already some evidence to suggest that sputum periostin reflects biological processes that circulating periostin does not. For example, sputum (but not serum) periostin levels have been found to be elevated in patients with eosinophilic asthma characterized by high numbers of columnar epithelial cells in sputum, possibly reflecting a phenotype with altered bronchial epithelial status [24]. Sputum periostin also reflects fixed airway obstruction more closely than serum periostin, probably due to the involvement of periostin in airway remodeling [26, 27]. Pavlidis and colleagues report that serum periostin is a poor

predictor of T2-high asthma as determined by epithelial transcriptomic profile, and discuss that their unpublished findings suggest sputum periostin may be more promising for this purpose [5]. In the current investigation there were generally more significant associations between sputum periostin with clinical characteristics than for serum periostin. The effect of a 2-week oral corticosteroid intervention was examined in a subset of asthma patients and was found to significantly reduce sputum periostin levels to a greater degree than observed in matched serum samples. Furthermore, only sputum and not serum periostin was related to reduced lung function and only sputum periostin showed associations with sputum IL-13. Furthermore, when patients were divided according to sputum inflammatory profile, only sputum periostin showed different levels among the four profiles being lowest in paucigranulocytic and highest in mixed granulocytic asthma.

One disease of the airways in which periostin has been extensively studied and known to be involved in disease pathology is that of chronic rhinosinusitis (CRS), where approximately 20% of sufferers have co-morbid asthma [28–30]. Such CRS patients often develop a late-onset, poorly controlled, eosinophilic asthma and nasal polyps are common [30]. The clinical usefulness of the ELISA system described in the current investigation (Assay B) has recently been tested in a cohort of patients with CRS where multiple associations between sputum periostin with disease parameters were revealed [29]. Sputum periostin levels were detectable, and significantly highest in CRS with comorbid asthma compared to without. Levels also correlated with exhaled NO, $FEV_1$ and eosinophil numbers in sputum as well as in nasal polyp and sinus tissue [29].

## Conclusion

We demonstrate how the detection of sputum periostin may be improved by an ELISA system containing antibodies that detect smaller cleavage products of the periostin protein, as these are found to a greater degree in sputum compared to serum. This finding may be of value to future studies investigating the utility of sputum periostin levels as a biomarker of airway disease, which has the potential to overcome some of the confounding factors complicating the use of serum periostin. Since serum periostin was found to be relatively unsuccessful in identifying an asthma population that particularly benefit from anti-IL-13 treatment, interest in periostin as a phenotype-specific biomarker has been somewhat reduced [31]. However, new studies continue to emerge describing a role for periostin in the pathogenesis of asthma [23], and serum periostin is also being used successfully in combination with other type two biomarkers such as blood eosinophils and FeNO to identify type-2 asthma [32]. As discussed, the proximity of sputum to the lung itself may be an important advantage of sputum periostin and as we show promising associations between sputum periostin with both sputum IL-13 and eosinophil levels, as well as reduced lung function, we believe it is worth pursuing further investigations into whether sputum periostin levels may hold more disease-relevant information than serum periostin.

## Supporting information

**S1 File. Inclusivity in global research.**
(DOCX)

**S1 Table. Assay validation for Assay A and Assay B.**
(DOCX)

**S2 Table. Linearity and recovery testing of assay A and assay B.**
(DOCX)

**S3 Table. Patient characteristics of cohort 2.**
(DOCX)

**S4 Table. Detection levels of Assay A and Assay B at follow-up visits.**
(DOCX)

**S5 Table. Correlations between serum and sputum periostin with sputum cytokines in cohort 2.**
(DOCX)

**S1 Fig. Typical calibration curves of assay A and assay B.**
(TIF)

**S2 Fig. Amino acid sequences of cleaved periostin.** The peptides detected by MALDI/TOF-MS are displayed by arrows. The locations of Asn23 and Lys391 are shown. The 37kDa periostin product was purified from human serum and subjected to MALDI/TOF-MS analysis for further identification. The peptide sequence of the band corresponding to 37 kDa detected by MALDI/TOF-MS analysis covered the amino acid sequence from Asn23 to Leu391 of periostin. The predicted molecular weight of the peptide from Asn23 to Leu391 is 40,141, compatible with the migrated band at 37 kDa.
(TIF)

**S3 Fig. Effects of MMP-7 on digestion of periostin, and periostin values obtained using Assay A and Assay B.** Recombinant periostin was incubated with 1 μg/mL of MMP-7 for 6 hours in the presence or absence of 1 mM EDTA. A Western blot showing periostin staining (A) and periostin values estimated by Assay A and Assay B (B) is depicted.
(TIF)

**S4 Fig. Correlation of periostin values estimated by Assay A and Assay B.** Serum periostin values were estimated in asthma patients using Assay A and Assay B.
(TIF)

**S5 Fig. Scatter plots showing all significant Spearman Rank correlations between sputum periostin and clinical characteristics (Table 2 in main article).** Plots show log sputum periostin levels measured by Assay B in patients with asthma from the BIOAIR study. Undetectable sputum periostin values are therefore not included in this analysis. Each plot is fitted with a linear regression line, the slope of which deviated significantly from zero ($p < 0.01$) for all parameters shown apart from FeNO.
(TIF)

**S1 Raw images.**
(PDF)

## Acknowledgments

We would like to thank members of the BIOAIR consortium: Lars I. Andersson, Maciej Kupczyk, Barbro Dahlén, Mina Gaga, Nikos M. Siafakas, Alberto Papi, Bianca Beghe, Guy Joos, Klaus F. Rabe, Peter J. Sterk, Elisabeth H. Bel, Sebastian L. Johnston, Pascal Chanez, Mark Gjomarkaj, Peter H. Howarth, Ewa Niżankowska-Mogilnicka and Roelinde Middelveld. We would also like to acknowledge the ChAMP (Centre for Allergy Research Highlights Markers of Asthma Phenotype) consortium which is funded by the Karolinska Institutet, AstraZeneca & Science for Life Laboratory Joint Research Collaboration. Finally, we would also like to thank Manali Mukherjee and Katherine Radford for their assistance with the collection of clinical samples.

## Author Contributions

**Conceptualization:** Shoichiro Ohta, Parameswaran Nair, Kenji Izuhara, Sven-Erik Dahlén, Anna James.

**Data curation:** Junya Ono.

**Funding acquisition:** Sven-Erik Dahlén.

**Investigation:** Junya Ono, Masayuki Takai, Ayami Kamei, Shoichiro Ohta, Kenji Izuhara.

**Methodology:** Junya Ono, Masayuki Takai, Ayami Kamei, Shoichiro Ohta, Kenji Izuhara, Anna James.

**Project administration:** Parameswaran Nair, Sven-Erik Dahlén.

**Resources:** Kenji Izuhara.

**Software:** Anna James.

**Supervision:** Parameswaran Nair, Kenji Izuhara, Sven-Erik Dahlén.

**Visualization:** Junya Ono.

**Writing – original draft:** Junya Ono, Parameswaran Nair, Anna James.

**Writing – review & editing:** Junya Ono, Masayuki Takai, Ayami Kamei, Shoichiro Ohta, Parameswaran Nair, Kenji Izuhara, Sven-Erik Dahlén, Anna James.

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
