## [Decision Letter · Decision Letter 0]

10 May 2022

PONE-D-22-03930A novel assay for improved detection of sputum periostin in patients with asthmaPLOS ONE

Dear Dr. Anna James,

Thank you for submitting your manuscript to PLOS ONE. After careful consideration, we feel that it has merit but does not fully meet PLOS ONE’s publication criteria as it currently stands. Therefore, we invite you to submit a revised version of the manuscript that addresses the points raised during the review process.

The manuscript is interesting but it needs additional improvement following the reviewers' suggestions.

We look forward to receiving your revised manuscript.

Kind regards,

Fabio Luigi Massimo Ricciardolo

Academic Editor

PLOS ONE

Journal Requirements:

In your cover letter, please note whether your blot/gel image data are in Supporting Information or posted at a public data repository, provide the repository URL if relevant, and provide specific details as to which raw blot/gel images, if any, are not available. Email us at plosone@plos.org if you have any questions

3. Please include a complete copy of PLOS’ questionnaire on inclusivity in global research in your revised manuscript. Our policy for research in this area aims to improve transparency in the reporting of research performed outside of researchers’ own country or community. The policy applies to researchers who have travelled to a different country to conduct research, research with Indigenous populations or their lands, and research on cultural artefacts. The questionnaire can also be requested at the journal’s discretion for any other submissions, even if these conditions are not met.  Please find more information on the policy and a link to download a blank copy of the questionnaire here: https://journals.plos.org/plosone/s/best-practices-in-research-reporting. Please upload a completed version of your questionnaire as Supporting Information when you resubmit your manuscript.

Additional Editor Comments (if provided):

The manuscript is interesting but it needs additional improvement following reviewers suggestions.

Reviewers' comments:

Reviewer's Responses to Questions

**Comments to the Author**

1. Is the manuscript technically sound, and do the data support the conclusions?

Reviewer #1: Partly

Reviewer #2: Partly

2. Has the statistical analysis been performed appropriately and rigorously? 

Reviewer #1: Yes

Reviewer #2: Yes

3. Have the authors made all data underlying the findings in their manuscript fully available?

Reviewer #1: Yes

Reviewer #2: Yes

4. Is the manuscript presented in an intelligible fashion and written in standard English?

Reviewer #1: No

Reviewer #2: Yes

5. Review Comments to the Author

Reviewer #1: The authors tested two ELISA assays that could be used to detect either whole or cleaved serum and/or sputum periostin in BIOAIR population. Furthermore, this study aimed to reveal the appropriateness of the sputum periostin as a biomarker in airway disease.

The paper is of interest, however, several issues need to be resolved.

Major comments:

1) In my opinion, the Authors explored several aspects related to periostin assessment. I think that they should focus the study on a singular point and explore it more deeply (i.e. validation of the Assay B as a feasible tool for the detection of cleaved sputum periostin). In this way, the article will be clear.

2) In my opinion the Material and Methods section must be rewritten. For example, in this section, the Immunoprecipitation and Western blotting paragraph can be put together with other paragraphs. Furthermore, I suggested moving all the figures and tables cited in the Result section.

3) The authors did not indicate in the Statistical Analysis paragraph for what analysis the author used the Chi2 test.

Minor comments:

1) Table 1 and 2 abbreviation: IL, interleukin this abbreviation, is not necessary.

2) Please insert in Figure 2 (panel A) the Spearman correlation rank and P-value.

3) In Table 1 the authors did not indicate the inhaled corticosteroid considered for the analyses (beclomethasone or fluticasone). Please provide this information.

4) I suggest changing the word “Atopic” with “Atopy” in Table 1.

Reviewer #2: The paper of Ono and colleagues compared two ELISA methods, assay A and Asssay B, and showed a higher efficacy of the latter for the detection of sputum periostin in samples from asthmatic patients. The Authors showed also a greater usefulness of sputum periostin as a biomarker of airway inflammation than serum periostin.

Despite the paper gives interesting information, it should be improved as it takes into account several facets which deserve a deeper investigation.

The authors showed different sputum periostin levels according to different asthma inflammatory phenotypes, being highest in mixed granulocytic asthma. I think it is interesting to assess whether sputum periostin correlates with inflammatory cytokines other than Type-2 mediators (i.e. IL-8, IL-17A/F). It is not clear what test was used to measure serum periostin.

Have the authors performed the ELISA tests on the whole sputum?

6. PLOS authors have the option to publish the peer review history of their article (what does this mean?). If published, this will include your full peer review and any attached files.

Reviewer #1: No

Reviewer #2: No

---

## [Author Response · Author response to Decision Letter 0]

8 Nov 2022

Dear PLOS One Editors,

We thank the reviewers for their helpful feedback and guidance, and we are pleased to resubmit our manuscript entitled “A novel assay for improved detection of sputum periostin in patients with asthma” for consideration for publication in PLOS One, along with our responses to comments from the editors and reviewers below.

As described in more detail below, our original plan was to add more data regarding sputum cytokines as suggested by reviewer 2, before our extended deadline of 11th October 2022. However, for reasons beyond our control we have not been able to do this. We have however addressed all other suggestions made by the reviewers including the addition of important new data relating to the clinical utility of sputum periostin as a potential biomarker in asthma.

Your sincerely,

Anna James, corresponding author

Editors’ comments

1.PLOS ONE now requires that authors provide the original uncropped and unadjusted images underlying all blot or gel results reported in a submission’s figures or Supporting Information files.

Response: Original blot or gel results have now been uploaded as a Supporting Information file. In order to meet the PLOS One requirements for the presentation of gel/blot results we have also slightly adjusted the figures which contain such data (figs 1, 5 and S3).

2. Please include a complete copy of PLOS’ questionnaire on inclusivity in global research in your revised manuscript.

Response: A completed version of the “Inclusivity in global research questionnaire” has been uploaded as Supporting Information.

Response: The funders had no role in study design, data collection and analysis, decision to publish, or preparation of the manuscript.

4. Thank you for stating the following in the Competing Interests section: [I have read the journal's policy and the authors of this manuscript have the following competing interests: JO, MT and AK are employees of Shino-Test Corporation. PN reports grants and/or personal fees from AstraZeneca, Teva, Sanofi, BI, Methapharm, Cyclomedica, GSK, Equillium, Foresee and Knopp. KI reports grants from Shino-test Corporation, AstraZeneca and Sanofi. SED reports personal fees from AstraZeneca, Cayman Chemicals, GSK, Novartis, Sanofi-Regeneron and TEVA. SO and AJ have no relevant financial disclosures.]. 

Please confirm that this does not alter your adherence to all PLOS ONE policies on sharing data and materials, by including the following statement: "This does not alter our adherence to PLOS ONE policies on sharing data and materials.” Please include your updated Competing Interests statement in your cover letter; we will change the online submission form on your behalf.

Response: This does not alter our adherence to PLOS ONE policies on sharing data and materials.

Reviewers’ comments

Reviewer #1: The authors tested two ELISA assays that could be used to detect either whole or cleaved serum and/or sputum periostin in BIOAIR population. Furthermore, this study aimed to reveal the appropriateness of the sputum periostin as a biomarker in airway disease.

The paper is of interest, however, several issues need to be resolved.

Major comments:

1. In my opinion, the Authors explored several aspects related to periostin assessment. I think that they should focus the study on a singular point and explore it more deeply (i.e. validation of the Assay B as a feasible tool for the detection of cleaved sputum periostin). In this way, the article will be clear. 

We fully agree with reviewer 1 and in order to improve the clarity of our manuscript we have therefore made the following changes:

1. The first paragraph of the Results section concerning development of the methodology used and entitled “Development of ELISAs detecting whole and cleaved periostin” has been moved to the Methods section and shortened somewhat. This will hopefully improve the focus of the results section which now describes associations between the sputum periostin levels measured, asthma and airway inflammation (line 97-166).

2. To further validate and investigate the clinical relevance of sputum periostin levels measured using assay B, as suggested, we have added new data regarding the effects of anti-inflammatory glucocorticoid therapy, an important factor often overlooked when novel biomarkers are proposed (lines 44, 85, 171, 262-272, 368). In addition we have added information regarding assay performance at repeat visits (lines 173, 209).

2. In my opinion the Material and Methods section must be rewritten. For example, in this section, the Immunoprecipitation and Western blotting paragraph can be put together with other paragraphs. Furthermore, I suggested moving all the figures and tables cited in the Result section. 

The Material and Methods section has now been adjusted as suggested. The Immunoprecipitation and Western blotting paragraph has been merged with the first paragraph on antibodies (line 96). Furthermore, and as described above in point 1, the first paragraph of the Results section along with all figures and tables concerning development of the methodology used have been moved to the Methods section to improve the distinction between the two aspects of this manuscript, firstly the development of a new method and secondly its validation in well-characterised patients with asthma. 

3. The authors did not indicate in the Statistical Analysis paragraph for what analysis the author used the Chi2 test. 

This Chi2 test was used to compare percentages of males/females, those taking oral corticosteroids and those with a positive skin prick test in the table showing patient characteristics. This has been clarified in the Statistics section line 196.

Minor comments:

1) Table 1 and 2 abbreviation: IL, interleukin this abbreviation, is not necessary. This abbreviation has been removed.

2) Please insert in Figure 2 (panel A) the Spearman correlation rank and P-value. The figure actually shows a linear regression, but we have added more information both about the Spearman correlation rank and P-value, as well as the linear regression correlation coefficient and associated p value, into the figure legend (line 238) as well as on figure 2.

3) In Table 1 the authors did not indicate the inhaled corticosteroid considered for the analyses (beclomethasone or fluticasone). Please provide this information. This information has been provided (table at line 177).

4) I suggest changing the word “Atopic” with “Atopy” in Table 1. This has been changed (table at line 177).

Reviewer #2: The paper of Ono and colleagues compared two ELISA methods, assay A and Asssay B, and showed a higher efficacy of the latter for the detection of sputum periostin in samples from asthmatic patients. The Authors showed also a greater usefulness of sputum periostin as a biomarker of airway inflammation than serum periostin. Despite the paper gives interesting information, it should be improved as it takes into account several facets which deserve a deeper investigation.

The authors showed different sputum periostin levels according to different asthma inflammatory phenotypes, being highest in mixed granulocytic asthma. I think it is interesting to assess whether sputum periostin correlates with inflammatory cytokines other than Type-2 mediators (i.e. IL-8, IL-17A/F). It is not clear what test was used to measure serum periostin. 

Regarding the last point, we have now added more information to describe how serum periostin was analysed in the current investigation, line 123-125. Serum periostin levels measured using this method (assay A) have been extensively reported in the literature (see reference 4 in manuscript for summary, Izuhara et al.).

We fully agree that it would be of interest to examine how sputum periostin relates to other inflammatory cytokines and began to plan for such an investigation after receiving this relevant comment. We decided to extract sputum and matched plasma samples from our biobanks where samples from the BIOAIR study are stored, and perform an OLINK multiplex analysis of 96 inflammatory mediators in these samples which would enable us to answer several important questions including those raised by reviewer #2, but also comparisons between local and circulating biomarker levels. However, due to general delays caused by the COVID pandemic, the biobank and waiting times for OLINK analysis at our core facility, we will not be able to add this information before the 6-month resubmission deadline at PLOS One (which led us to enquire about an extension). As this investigation has already been scheduled, the best we can do is follow up the current, initial investigation with a later publication, where we delve more deeply into the inflammatory pathways that relate to sputum periostin levels.

Have the authors performed the ELISA tests on the whole sputum? 

The ELISA tests were performed on sputum supernatants, the fraction containing soluble mediators after the cell pellet has been separated away in the process of sputum handling and this has been clarified in the text (line 190).

---

## [Decision Letter · Decision Letter 1]

23 Jan 2023

A novel assay for improved detection of sputum periostin in patients with asthma

PONE-D-22-03930R1

Dear Dr. James,

We’re pleased to inform you that your manuscript has been judged scientifically suitable for publication and will be formally accepted for publication once it meets all outstanding technical requirements.

Kind regards,

Shama Ahmad, Ph.D.

Academic Editor

PLOS ONE

Additional Editor Comments (optional):

Reviewers' comments:

Reviewer's Responses to Questions

**Comments to the Author**

1. If the authors have adequately addressed your comments raised in a previous round of review and you feel that this manuscript is now acceptable for publication, you may indicate that here to bypass the “Comments to the Author” section, enter your conflict of interest statement in the “Confidential to Editor” section, and submit your "Accept" recommendation.

Reviewer #1: All comments have been addressed

Reviewer #2: All comments have been addressed

2. Is the manuscript technically sound, and do the data support the conclusions?

Reviewer #1: Yes

Reviewer #2: Yes

3. Has the statistical analysis been performed appropriately and rigorously? 

Reviewer #1: Yes

Reviewer #2: Yes

4. Have the authors made all data underlying the findings in their manuscript fully available?

Reviewer #1: Yes

Reviewer #2: Yes

5. Is the manuscript presented in an intelligible fashion and written in standard English?

Reviewer #1: Yes

Reviewer #2: Yes

6. Review Comments to the Author

Reviewer #1: The authors replied exhaustively to the Reviewer's comments and is now considered to be suitable for publication in Plos One.

Reviewer #2: The authors amended the manuscript appropriately and provided satifactory answers. Themanuscript is suitable for publication.

7. PLOS authors have the option to publish the peer review history of their article (what does this mean?). If published, this will include your full peer review and any attached files.

Reviewer #1: No

Reviewer #2: No

---

## [Editor Report · Acceptance letter]

2 Feb 2023

PONE-D-22-03930R1 

A novel assay for improved detection of sputum periostin in patients with asthma 

Dear Dr. James:

I'm pleased to inform you that your manuscript has been deemed suitable for publication in PLOS ONE. Congratulations! Your manuscript is now with our production department. 

Kind regards, 

on behalf of

Dr. Shama Ahmad 

Academic Editor

PLOS ONE